# Studying the Oncolytic Activity of *Streptococcus pyogenes* Strains Against Hepatoma, Glioma, and Pancreatic Cancer *In Vitro* and *In Vivo*

**DOI:** 10.3390/microorganisms13010076

**Published:** 2025-01-03

**Authors:** Anna N. Tsapieva, Alexander N. Chernov, Nadezhda V. Duplik, Anastasiya O. Morozova, Tatiana A. Filatenkova, Mariia A. Suvorova, Elena Egidarova, Elvira S. Galimova, Kseniya Bogatireva, Alexander N. Suvorov

**Affiliations:** 1Scientific and Educational Center, Molecular Bases of Interaction of Microorganisms and Human of the Center for Personalized Medicine of Federal State Budgetary Scientific Institution, Institute of Experimental Medicine, Acad. Pavlov Street, 12, 197022 Saint Petersburg, Russia; nadezhdaduplik@gmail.com (N.V.D.); aomorozova1993@gmail.com (A.O.M.); lero269@gmail.com (T.A.F.); lena.egidarova.97@mail.ru (E.E.); elvira8galimova@gmail.com (E.S.G.); bogat.ksenia.p@gmail.com (K.B.); alexander_suvorov1@hotmail.com (A.N.S.); 2Department of Biological Chemistry, Federal State Budgetary Educational Institution of Higher Education Saint Petersburg, State Pediatric Medical University of the Ministry of Health of Russia, 194100 Saint Petersburg, Russia; 3Department of Biotechnology and Biomedicine, Disease Systems Immunology, Technical University of Denmark, Søltofts Plads 224, 2800 Kongens Lyngby, Denmark; marsuv@dtu.dk; 4Sechenov Institute of Evolutionary Physiology and Biochemistry of the Russian Academy of Sciences, Thorez av., 44, 194223 Saint Petersburg, Russia

**Keywords:** *Streptococcus pyogenes*, recombinant M protein, human glioma U251, pancreatic cancer PANC02, mice hepatoma 22a, cytotoxicity, mice survival rate

## Abstract

Background: Cancer remains a leading cause of mortality globally. Conventional treatment modalities, including radiation and chemotherapy, often fall short of achieving complete remission, highlighting the critical need for novel therapeutic strategies. One promising approach involves the oncolytic potential of Group A *Streptococcus* (GAS) strains for tumor treatment. This study aimed to investigate the oncolytic efficacy of *S. pyogenes* GUR and its M protein knockout mutant, *S. pyogenes* strain GURSA1, which was genetically constructed to minimize overall toxicity, against mouse hepatoma 22A, pancreatic cancer PANC02, and human glioma U251 cells, both *in vitro* and *in vivo*, using the C57BL/6 mouse model. Methods: The *in vitro* oncolytic cytotoxic activity of GAS strains was studied against human glioma U251, pancreatic cancer PANC02, murine hepatoma 22a, and normal skin fibroblast cells using the MTT assay and the real-time xCELLigence system. A syngeneic mouse model of hepatoma and pancreatic cancer was used to evaluate the *in vivo* oncolytic effect of GAS strains. Statistical analysis was conducted using Student’s *t*-test and Mann–Whitney U-test with GraphPad Prism software. Results: The *in vitro* model showed that the live *S. pyogenes* GUR strain had a strong cytotoxic effect (67.4 ± 1.9%) against pancreatic cancer PANC02 cells. This strain exhibited moderate (38.0 ± 1.8%) and weak (16.3 ± 5.4%) oncolytic activities against glioma and hepatoma cells, respectively. In contrast, the *S. pyogenes* GURSA1 strain demonstrated strong (86.5 ± 1.6%) and moderate (36.5 ± 1.8%) oncolytic activities against glioma and hepatoma cells. Additionally, the *S. pyogenes* GURSA1 strain did not exhibit cytotoxic activity against healthy skin fibroblast cells (cell viability 104.2 ± 1.3%, *p =* 0.2542). We demonstrated that tumor treatment with *S. pyogenes* GURSA1 significantly increased the lifespan of C57BL/6 mice with hepatoma (34 days, *p =* 0.040) and pancreatic cancer (32 days, *p =* 0.039) relative to the control groups (24 and 28 days, respectively). Increased lifespan was accompanied by a slowdown in tumor progression, as evidenced by a reduction in the growth of hepatoma and pancreatic cancer tumors under treatment with GAS strains in mice. Conclusions: Both *S. pyogenes* GUR and *S. pyogenes* GURSA1 strains demonstrated strong oncolytic activity against murine hepatoma 22a, pancreatic cancer PANC02, and human U251 glioma cells *in vitro*. In contrast, *S. pyogenes* GUR and GURSA1 did not show toxicity against human normal skin fibroblasts. The overall survival rate and lifespan of mice treated with *S. pyogenes* GURSA1, a strain lacking the M protein on its surface, were significantly higher compared to the control and *S. pyogenes* GUR strain groups.

## 1. Introduction

It is obvious that in recent years significant progress has been achieved in the field of cancer therapy; however, it continues to be one of the most common causes of death in developed countries. According to the forecasts of the World Health Organization (WHO), by 2040 the number of oncological diseases in the world will increase to 29.5 million people compared to 18.1 million in 2018 [1,2]. Conventional methods, such as radiation and chemotherapy, do not provide a complete cure, which necessitates the search for new alternative approaches, one of which is the tumors’ treatment using microorganisms.

The study and use of bacterial strains with the ability to inhibit cancer development has a fairly long history. Even long before radiation and chemotherapy, practicing physicians noticed that spontaneous healing of cancer cases often occurred against the background of a bacterial or viral infection accompanied by fever. Largely with the participation of research by American surgeon William Coley, it was proved that a number of bacteria, primarily *Streptococci*, have pronounced oncolytic properties. During his research, Coley managed to treat more than 1500 terminal patients with various cancer types with a high success rate [3,4].

The ability to cure cancer or make it a treatable chronic disease is a force that drives discovery and innovation in oncological microbial therapy. Bacterial-mediated cancer therapy (BMCT) may be considered as a new direction of antitumor therapy. Microbial therapy has the potential to solve many clinical problems that cannot be solved with current oncological methods. For example, BMCT may be efficient for the treatment of resistant metastatic cancers, and for diminishing the immunosuppressive tumor cells’ effects. A significant advantage of microbial therapy is its specific effect on cancer cells and tissues. Thus, microbial therapy is well suited for the treatment of metastatic diseases, the main death cause from cancer [5]. It was reported that strains of *E. coli*, *Salmonella* sp., *Bifidobacterium* sp., *Pseudomonas aeruginosa*, *Clostridium* sp., and *Streptococcus* sp. are able to selectively migrate to and colonize the tumor hypoxic niche [6,7,8]. In addition, these microorganisms can be genetically modified to reduce systemic toxicity, increase anticancer efficacy, modify the expression of genes and proteins in cancer cells, or detect and directly treat cancer [8,9]. To date, many preclinical studies on bacterial microbial therapy have shown a slowdown in cancer growth and an increase in animal survival [8,10,11]. A complete cancer regression was achieved with oncolytic bacteria in immunocompetent animals with syngeneic cancers and in companion dogs with spontaneous tumors [12,13,14]. Attenuated live bacteria have satisfactory safety profiles in both healthy and tumor-bearing animals [15,16].

Despite the successful use of bacteria, including group A *Streptococci* (GAS), for cancer treatment, the anticancer mechanisms of oncolytic bacteria are still not well understood [17]. These bacteria consume nutrients and secrete toxins and cytolytic agents (such as lipases and proteases), inducing apoptosis or autophagy in cancer cells [18]. It was shown that some streptococcal proteins, in particular, the enzymes arginine deiminase or streptokinase, can significantly slow down tumor cells’ proliferation [19,20,21]. OK-432 (picibanil), a drug based on *S. pyogenes*, induces the death of lymphangioma cells [22]. Thus, GAS strains represent promising candidates for investigating cytotoxic mechanisms and effects against cancer cells, as well as for developing cellular vaccines and innovative approaches to anticancer therapy. Our preliminary studies on the oncolytic activity of *S. pyogenes* GUR and GURSA1 across a wide panel of cancer cell lines demonstrated strong efficacy against brain tumors, sarcoma, hepatoma, and pancreatic cancer cells. Given that the oncolytic activity of *S. pyogenes* against sarcoma was well-documented in previous studies, dating back to the work of W. Coley, we decided to focus our efforts on a more detailed investigation in the glioma, hepatoma, and pancreatic cancer models.

The current study explores the potential of oncolytic therapy using GAS strains, with a particular focus on comparing the effects of the original *S. pyogenes* GUR strain and its M protein knockout mutant, *S. pyogenes* GURSA1, in syngeneic mouse models of hepatoma and pancreatic carcinoma. The key novelty of this research lies in examining the distinct therapeutic effects between the wild-type strain and the knockout mutant.

## 2. Materials and Methods

### 2.1. Streptococcus pyogenes Strains

For this study, four GAS strains with previously discovered oncolytic activities were chosen. *S. pyogenes* GUR (type emm111) is a throat isolate from a scarlet fever patient, which was used clinically to treat cancer patients in the former Soviet Union for more than 20 years [23]. *S. pyogenes* GUR was kindly provided by prof. Chereshnev V.A., Perm State University (Perm, Russia). *S. pyogenes* GURSA1 is a derivative of the *S. pyogenes* GUR strain with an inactivated M protein gene [21]. *S. pyogenes* GUR and *S. pyogenes* GURSA1 strains showed cytotoxic activity against murine cancer cells [19]. The strains were cultivated in Todd-Hewitt broth (Condalab, Spain) for 16 h at 37 °C from single colonies. For all experiments, 10^6^ CFU of *S. pyogenes* in Dulbecco’s modified eagle medium (DMEM) or PBS was used. The optical density of overnight cultures was measured at 600 nm to estimate the bacterial cell count, based on previously established calibration curves. The cultures were then centrifuged at 6000× *g* for 5 min, washed once with PBS, and resuspended in DMEM or PBS to achieve a uniform bacterial concentration.

### 2.2. Obtaining M Protein Gene Knockout Mutant of S. pyogenes GUR

To generate a knockout of the M protein gene (*emm*) in *S. pyogenes* GUR, we designed an integrative plasmid for homologous recombination. The plasmid was constructed using the pT7ermB vector, which contains part of the *emm* gene necessary for integration into the *S. pyogenes* GUR chromosome. By using modified primers MF_Hind (acagcagaggcttaggcggaggatcattg) and MR_EcoR (atttcttaagatgctgccatagctt), the *emm* gene fragment was successfully amplified. The amplicons and the pT7ermB vector were digested with HindIII and EcoRI, and then ligated into the pT7ermB (*emm111*) vector. The resulting vector was used to transform *S. pyogenes* GUR. *S. pyogenes* GUR was grown to mid-log phase in Todd-Hewitt (TH) broth (Condalab, Spain). The cells were made electrocompetent by washing twice with ice-cold 10% glycerol and resuspending them in 1/40 (by volume) of the same solution. Approximately 1 µg of the integrative (nonreplicative) plasmid was electroporated into the competent *S. pyogenes* cells using a standard electroporation protocol (1.8 kV, 25 µF, 200 Ω). Following electroporation, the cells were recovered in 1 mL of TH broth and incubated at 37 °C for 1 h to allow for plasmid integration in the bacterial chromosome. After recovery, the transformed cells were plated on selective media containing 2.5 µg/mL erythromycin to select for the integrants. Recombinants were allowed to grow for 24–48 h at 37 °C. Colonies that appeared on the selective media were picked and grown in TH broth with antibiotic selection for further analysis. To confirm the successful integration of the plasmid and knockout of the M protein gene, PCR confirmation were conducted using primers *ERM1* (gggcccaaaatttgtttga), *ERM2* (tcggcagcgactcatagaat), and *EMM* (agtcgtaggagcagggttagc).

### 2.3. Cell Cultures

Human glioma U251, pancreas cancer PANC-02, and murine hepatoma MH-22a were obtained from the Russian collection of cell cultures (Saint-Petersburg, Russia). These cells were cultured at 1.0 × 10^6^/mL in 25 cm^2^ flasks (T25 Nunclon, ThermoFisher Scientific Inc., Waltham, MA, USA) in RPMI-1640 and DMEM (Sigma-Aldrich, Saint Louis, MO, USA) containing 10% fetal bovine serum (FBS, Sigma-Aldrich, USA), and 2 mM glutamine (Sigma-Aldrich, USA) at 37 °C, 95% humidity, and 5% CO_2_ for 1–2 days [24,25]. Normal human skin fibroblasts (NHF) were isolated following an aseptic surgical procedure from a donor. The cells were obtained according to standardized methodology [26]. Approximately 1 × 10^6^ cells were seeded into an uncoated 25 cm^2^ tissue culture flask in DMEM with 10% FBS and incubated at 37 °C in a 5% CO_2_ atmosphere and 95% humidity. Subculture was performed when cells reached 70–90% confluence.

### 2.4. MTT Assay

The oncolytic activity of GAS against cancer cell lines and normal skin fibroblast cells was assessed using the MTT assay [27,28]. For this purpose, 1 × 10^4^ cells/well, suspended in 50 μL of RPMI-1640 or DMEM with 10% FBS, were seeded into the wells of 96-well flat-bottom plates (TPP, Trasadingen, Switzerland) 1 day before GAS treatment and incubated for 24 h at 37 °C in a 5% CO_2_ atmosphere. The following day, 10 μL of the GAS samples (*GUR*, *GURSA1*, and strains 7, 21 at 1.2 × 10^6^ and 2.5 × 10^6^ cells/mL) were added to 90 μL of the medium in each well. Three replicates were performed for each GAS concentration. As a positive control, 100 μL of DMEM was added to the wells containing tumor cells instead of GAS. Negative controls were included, with 10 μL of Triton X-100 added to the cells or empty wells. The plates were incubated for 20 h in a CO_2_ incubator. Then, 50 μL of resazurin dye was added to each well, and incubation continued until a color difference was observed between the positive and negative controls. The optical density (OD) of the solution in the wells was measured at a wavelength of 570 nm, subtracting the OD at 690 nm as the background, using a SpectraMax 250 plate reader (Molecular Devices, San Jose, CA, USA). The percentage of dead cells was calculated by comparing the OD of the samples with the positive (100% viable cells) and negative (0% viable cells) controls.

### 2.5. Real-Time Cytotoxicity Analysis Using RT xCELLigence System

For the real-time detection of GAS oncolytic activity against cancer cell lines and NHF cells, the RT xCELLigence system (Agilent Technologies Inc., Santa Clara, CA, USA) with E-Plate L8 was used [29]. The RT xCELLigence system records changes in the electronic impedance (resistance) of a specially designed plate containing embedded gold micro-electrodes. The change in resistance over time enables the generation of a graph showing the functional dependence of the cell index on time. On day 0, 150 µL of a medium and 150 µL of a cell suspension containing 50,000–200,000 cells per well (U251, PANC02, and MH-22a) were added to each well of the E-Plate L8. The method was standardized by measuring the OD of the cell suspension (OD = 0.200 ± 0.010) using a spectrophotometer at a wavelength of 600 nm. The loaded plates were incubated at 5% CO_2_ in the RT xCELLigence system for 24 h. After this period, overnight cultures of GAS strains (at 10^6^ CFU in 300 mL of DMEM) were added. Cancer cells were then cultured for an additional day, with the electrical resistance of the cell monolayer recorded every minute.

### 2.6. Hepatoma 22a and Pancreatic Cancer PANC02 Mouse Models

To establish a mouse hepatoma model, we used hepatoma 22A cells, which were harvested from the culture flask, resuspended in a physiological solution at a concentration of 10^5^ cells/mL, and subcutaneously inoculated into the right flank of C57BL/6 mice (*n* = 66, Rappolovo Nursery, Russia) under short-term ether anesthesia. The study used female C57BL/6 mice weighing 16–18 g. The animals were maintained under a 12 h light/12 h dark cycle and were provided with standard food (concentrated combined feed in the form of briquettes) and water ad libitum. Tumor size was measured 10 days post-injection, with an average size of 2.5 ± 0.5 mm^3^. The mice were then divided into three groups: GUR (*n* = 22) and GURSA1 (*n* = 22) groups received intratumoral injections of *S. pyogenes* GUR and *S. pyogenes* GURSA1 cells, respectively, at a dose of 10^6^ cells in 50 μL of suspension. The control group (*n* = 22) received no treatment.

Similarly, a C57BL/6 mouse model with PANC02 cancer cells was established according to the protocol described above. When the tumors reached a size of 5 ± 0.5 mm^3^ (18 days post-injection), the mice were divided into the same groups: GUR (*n* = 8), *GURSA1* (*n* = 8), and control (*n* = 7). The survival of all animals was monitored for 40 days. At the end of the experiment, the animals were euthanized by an overdose of ether anesthesia.

The ability of *S. pyogenes* GURSA1 to migrate into the tumor site following intraperitoneal administration was also studied using the PANC02 cancer C57BL/6 mouse model. When tumors had reached a size of 5 ± 0.5 mm^3^, one group (*n* = 6) was intraperitoneally infected with 10^6^ CFU of *S. pyogenes GURSA1* in a volume of 50 μL dissolved in PBS. Animals that received an equivalent volume of PBS alone served as controls (*n* = 6). Mice were sacrificed one day after the administration of *S. pyogenes* GURSA1. At the end of each experiment, tumor, spleen, and liver samples were homogenized in 500 μL of PBS using a vibrating ball mill (MM 400, Retsch Inc., Newtown, PA, USA), and tenfold serial dilutions were plated on Petri dishes containing Columbia blood agar (Conda, Spain). The seeded dishes were incubated at 37°C for 24 h. Colonies of microorganisms were identified using mass spectrometric analysis on a Bactoscreen device (Litech, Moscow, Russia).

All animal experiments were conducted in accordance with the “Rules of Laboratory Practice” (Ministry of Health of the Russian Federation, No. 708). The study was approved by the Local Ethics Committee for Animal Care and Use at the Institute of Experimental Medicine, Saint Petersburg, Russia (Protocol 3/23 dated 20 September 2023).

### 2.7. Acute Oral Toxicity

The acute oral toxicity test was performed using C57BL/6 mice, aged 8–10 weeks, weighing between 18.6 and 22.7 g. The mice were allowed free access to food and water, except for 4 h prior to the experiment. Four groups of female mice were divided by strain (*S. pyogenes* GUR and *S. pyogenes* GURSA) and administered doses (10^6^ or 10^8^ CFU/mouse). The GAS strains were administered intragastrically at a volume of 500 µL. Clinical signs, changes in body weight, and necropsy findings were investigated during the test period of 14 days.

### 2.8. Statistical Analysis

All experiments were performed in triplicate. Statistical significance between the means of different treatments and their respective control groups was determined using Student’s *t*-test. Data are presented as means ± standard deviation and were considered significant at *p <* 0.05. For comparisons between two independent groups with a small sample size (*n* < 30), the nonparametric Mann–Whitney U-test was used [30]. Descriptive statistics were performed using GraphPad Prism software (La Jolla, CA, USA, version 6.01, 21 September 2012).

## 3. Results

### 3.1. In Vitro Study of Oncolytic Effects of GUR and GURSA1 Strains Against Hepatoma 22a, Pancreatic Cancer PANC02, and U251 Glioma Cells

The M protein of *S. pyogenes* is a major virulence factor, responsible for multiple functions such as adherence to host cells, immune evasion, and promoting bacterial survival in the host. The gene encoding the M protein is part of the *emm* gene family, and disruption or knockout of this gene significantly impacts the pathogenicity of *S. pyogenes*. Generating an M protein gene knockout mutant allowed for the creation of a less virulent variant of *S. pyogenes* GUR, enabling the investigation of the role of this protein in the oncolytic activity of the *S. pyogenes* GUR strain. The M protein knockout mutant was successfully created according to the methodology described above and was named *S. pyogenes* GURSA1. The whole genome sequences of *S. pyogenes GUR* and *S. pyogenes GURSA1* are available from our previous work [31]. Interestingly, a genomic study of the *S. pyogenes* GUR strain, belonging to M111 serotype, revealed a stop codon in the other gene of *mga* regulon—*mrp*, making the entire regulon nonfunctional.

As the first step of the study, we tested the oncolytic activities of the *S. pyogenes* GUR and *S. pyogenes* GURSA1 strains using MTT assay against murine hepatoma 22a, pancreatic cancer PANC02, and human U251 glioma cells (Figure 1).

The results in Figure 1 show the viability of cancer cells under exposure to live GAS strains, as measured by changes in metabolic activity according to the MTT assay. A significant difference in activity was observed between the strains within each cancer cell line. Notably, different cell lines exhibited varying sensitivities to the strains. Neither of the two strains demonstrated broad-spectrum anticancer activity, but rather exhibited individual cytotoxicity against specific cancer cells. The *S. pyogenes* GUR strain exerts a strong cytotoxic effect against pancreatic cancer PANC02 cells, resulting in a decrease in metabolic activity by up to 67.4 ± 1.9%. However, against glioma and hepatoma cells, the GUR strain exhibited moderate (38.0 ± 1.8%) and weak (16.3 ± 5.4%) oncolytic activity, respectively. In contrast, the *S. pyogenes* GURSA1 strain exhibited strong (86.5 ± 1.6%) and moderate (36.5 ± 1.8%) oncolytic activity against glioma and hepatoma cells, respectively, but was ineffective against pancreatic cancer cells (5.5 ± 2.1%). However, the oncolytic activity of *S. pyogenes* GURSA1 was significantly stronger than that of *S. pyogenes* GUR against U251 glioma (*p* = 0.000004) and 22a hepatoma (*p* = 0.003625) cells. In contrast, the oncolytic activity of *S. pyogenes* GUR was significantly stronger (*p* = 0.000014) than that of *S. pyogenes* GURSA1 against PANC02 pancreatic cancer cells.

As the MTT assay measures cancer cell viability indirectly by assessing changes in metabolic activity, we also investigated the oncolytic activity of the *S. pyogenes* GUR and *S. pyogenes* GURSA1 strains against 22a hepatoma, PANC02 pancreatic cancer, and U251 glioma cells using the RT xCelligence system. The RT xCelligence system allows for the direct real-time observation of the number of viable cells during GAS exposure (Figure 2, Figure 3 and Figure 4).

The results obtained in these experiments confirmed the data from the MTT assay. The *S. pyogenes* GURSA1 strain exhibited higher cytotoxicity compared to the GUR strain against hepatoma and astrocytoma cells, while the GUR strain was more active against pancreatic cancer cells. The real-time activity results in Figure 2, Figure 3 and Figure 4 also show that *S. pyogenes* GUR exerted moderate to strong oncolytic effects against glioma (*p* = 0.0495), hepatoma (*p* < 0.01), and pancreatic cancer cells (*p* < 0.01). *S. pyogenes* GURSA1 exerted moderate to strong oncolytic effects against hepatoma (*p* < 0.05), pancreatic cancer (*p* < 0.05), and glioma cells (*p* = 0.0080), respectively.

The results in Figure 2 demonstrate the viability of hepatoma cells under exposure to live GAS strains while the strains were added at 24 h growth point. The graph shows that *S. pyogenes* GURSA1 has the ability to kill hepatoma cells already after 1 h of exposure. Nevertheless, although the *S. pyogenes* GURSA1 strain induced faster cell death, both strains resulted in the same rate of cancer cell mortality after just 6 h of exposure.

In the case of pancreatic cancer cells, *S. pyogenes* GUR (Figure 3) demonstrated a significantly faster cytotoxic effect, killing the cancer cells after just 2 h of exposure. In contrast, the *S. pyogenes* GURSA1 strain showed a much weaker effect on the hepatoma model, and even after 24 h of observation, it did not achieve the same level of impact as *S. pyogenes* GUR. But the cytotoxic effect of *S. pyogenes* GURSA1 on pancreatic cancer PANC02 was still significant (*p* < 0.05).

Figure 4 demonstrates the results of the influence of the two studied strains, which are very similar to those obtained with the PANC02 model. However, the strains behaved oppositely in this model. U251 astrocytoma cells were more sensitive to the effect of *S. pyogenes* GURSA1 and slightly more resistant to *S. pyogenes* GUR. Even after 24 h of observation, the cytotoxic activity of the strains against astrocytoma cells showed a notable difference.

According to the data obtained from these two *in vivo* models testing the antitumor activity of live strains, there is no direct correlation between cytotoxic activity and the ability of the *S. pyogenes* strain to express the M protein on its surface. Both studies demonstrated similar rates of anticancer activity for *S. pyogenes* GUR and *S. pyogenes* GURSA1 against certain cell lines, with activity being strain-specific.

### 3.2. In Vitro Study of Effects of S. pyogenes GUR and S. pyogenes GURSA1 Strains Against Normal Skin Fibroblast Cells

In the next step, we investigated the oncolytic activity of *S. pyogenes* GUR and *S. pyogenes* GURSA1 on healthy skin fibroblast cells using the MTT assay and RT xCELLigence system (Figure 5).

The results of the MTT assay in Figure 5 indicate that the *S. pyogenes* GUR strain has a weak cytotoxic effect (6.9 ± 3.8% cytotoxicity, cell viability 93.1 ± 3.7%, *p* < 0.05) on skin fibroblast cells. In contrast, the *S. pyogenes* GURSA1 strain does not exhibit cytotoxic activity (cell viability 104.2 ± 1.3%, *p* = 0.2542) against healthy fibroblast cells.

### 3.3. In Vivo Study of Oncolytic Effects of S. pyogenes GUR and GURSA1 Strains Against Hepatoma 22a and Pancreatic Cancer PANC02 Cells

We also investigated the oncolytic effects of the *S. pyogenes* GUR and GURSA1 strains on the overall survival of mice with hepatoma 22a and PANC02 pancreatic cancer (Figure 6).

The results in Figure 6 show that the inoculation of *S. pyogenes* GURSA1 significantly increased the lifespan of C57BL/6 mice with hepatoma (median survival 34 days) and pancreatic cancer (median survival 32 days) compared to the control groups (median survival 24 and 28 days, respectively). However, the application of *S. pyogenes* GUR shortened the lifespan of mice with both hepatoma (median survival 14 days) and pancreatic cancer (median survival 22 days) compared to the control group.

An increase in the lifespan of mice with cancers treated with the *S. pyogenes* GUR and *S. pyogenes* GURSA1 strains was accompanied by slower tumor growth of hepatoma 22A and PANC02 pancreatic cancer compared to the control group (Figure 7).

Although the treatment of tumors with *S. pyogenes* GUR led to a slowdown in tumor progression, which was more pronounced in the pancreatic cancer model, it did not result in an increase in the animals’ lifespans. Moreover, it led to a reduction in the average lifespan of mice in the *S. pyogenes* GUR treatment groups in both models. In contrast, treatment with *S. pyogenes* GURSA1 significantly prolonged the animals’ lifespans and inhibited tumor progression in both models.

Finally, using the C57BL/6 mice pancreatic cancer model, we studied the ability of *S. pyogenes* GURSA1 to migrate to the tumor site following intraperitoneal administration (Table 1). As a result, we demonstrated that the *S. pyogenes* GURSA1 strain selectively colonized tumors 24 h after intraperitoneal administration (10^6^ CFU/mouse). Notably, *S. pyogenes* GURSA1 was completely absent from both the spleen and liver of the mice, highlighting its selective tumor targeting

The ability of oncolytic bacteria to selectively accumulate in tumors without colonizing normal tissues and organs is one of their most important characteristics. This property enables the development of genetic constructs based on oncolytic bacteria for the targeted delivery of anticancer drugs to solid tumors.

### 3.4. Studying Acute Oral Toxicity of S. pyogenes GUR and GURSA1 Strains

Acute oral toxicity was assessed using a two-dose acute oral toxicity test. The doses were selected based on the potential therapeutic dose—10^6^ CFU per mouse—and a 100-fold higher dose, 10^8^ CFU per mouse. No fatalities or clinical signs were observed in the mice throughout the study. There were no significant changes in body weight in the mice administered *S. pyogenes* strains (Table 2). Furthermore, no abnormal findings were noted during necropsy. These results suggest that *S. pyogenes* GUR and *S. pyogenes* GURSA1 did not adversely affect the health of the mice.

## 4. Discussion

Most of the oncolytic bacteria discovered by now, *Klebsiella*, *Listeria*, *Mycobacteria*, *Proteus*, *Salmonella*, and *Clostridia*, belong to the group of pathogens capable of long persistence in the human microbiome. It might be possible that these potentially pathogenic bacteria evolved as an internal monitoring system, overseeing the host’s immune condition and tissue state for the appearance of cancer cells. Following this logic, *S. pyogenes*, which can asymptomatically colonize some individuals for decades, appears to be an ideal anticancer agent. *S. pyogenes*, being a microaerophilic bacterium, readily colonizes and propagates in fast-growing tumor tissue lacking appropriate vascularization, producing numerous enzymes and toxins that impede tumor development. It is not surprising that in the case of intraperitoneal injection of streptococci, none of the bacteria were found in the internal organs but were colonizing the tumor implant (Table 1). Another factor supporting the use of *S. pyogenes* as an anticancer tool is its strong pro-inflammatory activity, which induces a switch in macrophage response from the tumor-suppressive M2 mode to the M1 mode [32].

Using *in vitro* tools, we established that live *S. pyogenes* GUR and GURSA1 strains have a strong anticancer effect against human glioma U251, pancreatic cancer PANC02, and mouse hepatoma 22a cells by MTT assay and the RT xCelligence system. At the same time, it was determined that both strains exhibited low toxicity on human normal skin fibroblasts (Figure 5). An acute oral toxicity test also demonstrated low overall strain toxicity: the administration of *S. pyogenes* GUR and *S. pyogenes* GURSA1 even in a 100-fold therapeutic dose did not adversely affect the health of the mice. In our previous publication, we also demonstrated that some GAS strains, especially *S. pyogenes* GURSA1, stimulated the survival of human perifocal brain tissue cells [28].

It is known that the M protein is the main surface-associated virulence factor of GAS and a key antigen target of host immunity [31]. The previously created M protein knockout mutant *S. pyogenes* GURSA1 was expected to have reduced oncolytic properties, but also lower overall toxicity compared to the wild-type *S. pyogenes* GUR strain. Nevertheless, the *in vitro* experiments showed that both strains exhibited comparable cytotoxic activity against tumor cells, which varied in degree across different cell lines.

In contrast to our expectations, *in vivo* experiments demonstrated that *S. pyogenes* GURSA1, but not *S. pyogenes* GUR, significantly increased the lifespan of C57BL/6 mice with hepatoma (34 days, *p* = 0.040) and pancreatic cancer (32 days, *p* = 0.039), compared to the control groups (24 and 28 days, respectively). This effect was accompanied by a slowdown in tumor progression in both hepatoma and pancreatic cancer models. The application of *S. pyogenes* GUR also delayed tumor development in mice, but it shortened the animals’ lifespans compared to the control group in both tumor models.

As the next step, we plan to conduct clinical trials of *S. pyogenes* GURSA1 in patients with tumors of the pancreatobiliary zone to assess efficacy, administration routes, and safety. The major limitation of using streptococci as antitumor agents is their virulent nature, which is not surprising to any physician treating patients with sore throats. However, by making this bacterium deficient for the for the M protein fibrils on the surface, they immediately transformed into the bacterial prey for the macrophages and T cell killers. Another limitation of oncolytic bacterial therapy is the relatively narrow spectrum of action, as their activity is limited to certain types of cancer, and they may not be effective against all cancer cell types, which restricts their use as a broad-spectrum anticancer agent. The presence of immunity and rapid elimination from the body may also reduce their therapeutic significance. Also, the methods for eliminating bacteria from the body to control bacterial infection should be carefully considered before therapy, as well as the long-term consequences of it.

## 5. Conclusions

Our results show that both the *S. pyogenes* GUR and GURSA1 strains exhibit strong oncolytic activity against murine hepatoma 22a, pancreatic cancer PANC02, and human U251 glioma cells *in vitro*, with no toxicity observed against human normal skin fibroblasts. *In vivo*, the *S. pyogenes* GURSA1 strain significantly increased the lifespans of C57BL/6 mice with hepatoma (34 days, *p* = 0.040) and pancreatic cancer (32 days, *p* = 0.039), compared to the control groups (24 and 28 days, respectively). Additionally, *S. pyogenes* GURSA1 selectively colonized tumors 24 h after intraperitoneal administration, without detection in spleens or livers. These findings highlight the distinct oncolytic potential of *S. pyogenes* GURSA1 and its promising therapeutic effects. Further studies are needed to explore the role of the M protein in this activity and identify bacterial factors contributing to its oncolytic effects.

## Figures and Tables

**Figure 1 microorganisms-13-00076-f001:**
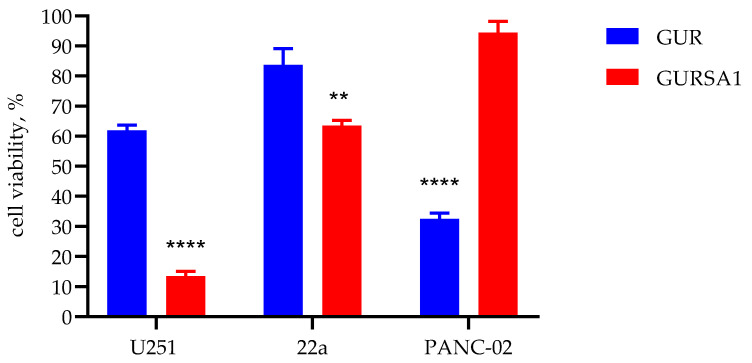
Oncolytic activity of live *S. pyogenes* GUR and GURSA1 against glioma U251, murine hepatoma 22a, and pancreatic cancer PANC02 cells using MTT assay; **, **** Statistically significant (*p* < 0.01, *p* < 0.0001) differences between activities of strains within each tumor cell type.

**Figure 2 microorganisms-13-00076-f002:**
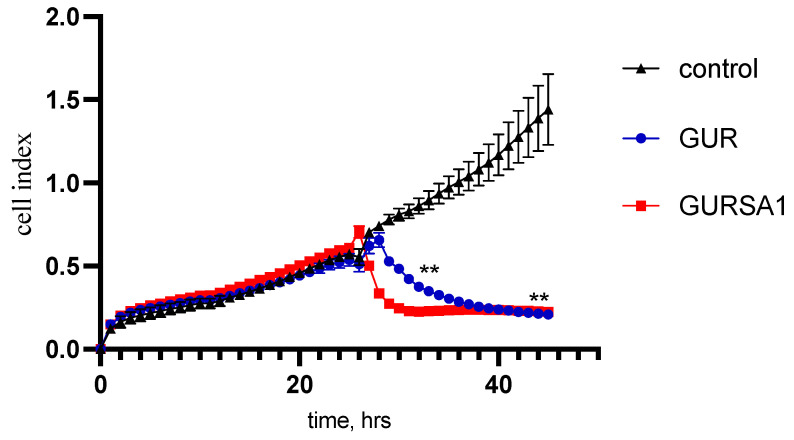
Oncolytic activity of *S. pyogenes* GUR and *S. pyogenes* GURSA1 against murine hepatoma 22a using RT xCELLigence system. ** Statistically significant (*p* < 0.01) difference between activities of strains and control.

**Figure 3 microorganisms-13-00076-f003:**
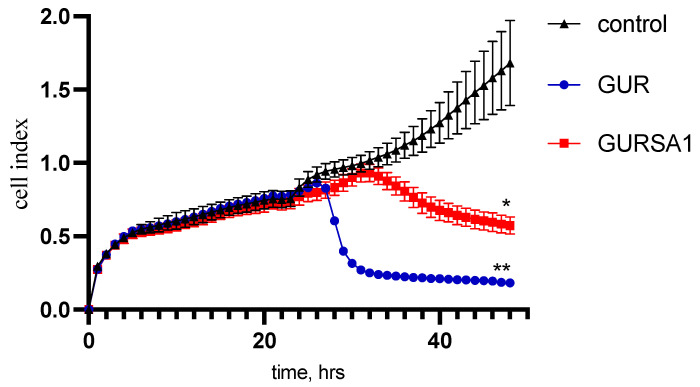
Oncolytic activity of *S. pyogenes* GUR and *S. pyogenes* GURSA1 against pancreatic cancer PANC02 using RT xCELLigence system. *, ** Statistically significant (*p* < 0.05, *p* < 0.01) differences between activities of strains and control.

**Figure 4 microorganisms-13-00076-f004:**
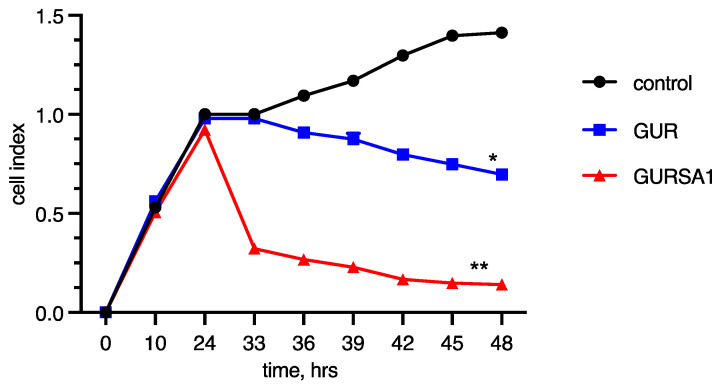
Oncolytic activity of *S. pyogenes* GUR and *S. pyogenes* GURSA1 against U251 glioma cells using RT xCELLigence system. *, ** Statistically significant (*p* < 0.05, *p* < 0.01) differences between activities of strains and control.

**Figure 5 microorganisms-13-00076-f005:**
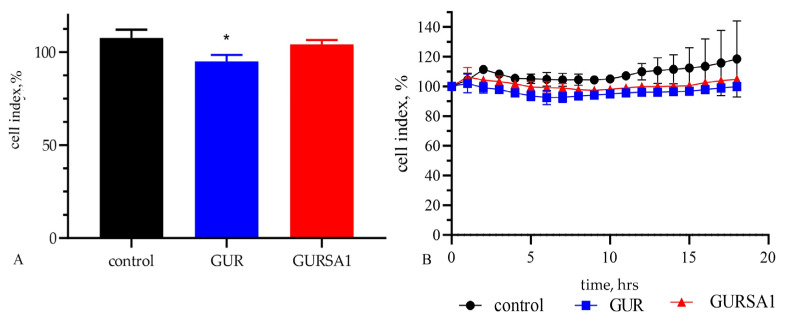
Oncolytic activity of live *S. pyogenes* GUR and *S. pyogenes* GURSA1 against healthy skin fibroblast cells using (**A**) MTT assay and (**B**) RT xCELLigence system. * Statistically significant (*p <* 0.05) difference between activity of *S. pyogenes* GUR and control.

**Figure 6 microorganisms-13-00076-f006:**
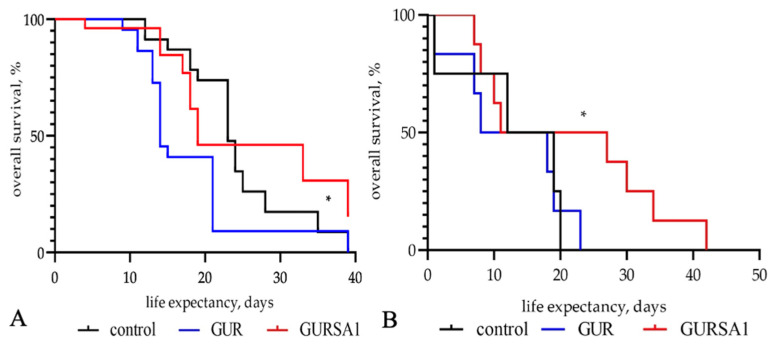
Influence of live *S. pyogenes* GUR and *S. pyogenes* GURSA1 strains on C57BL/6 mice overall survival with (**A**) murine hepatoma, and (**B**) pancreatic cancer. 1—control, 2—*S. pyogenes* GUR, and 3—S. pyogenes GURSA1. * Statistically significant (*p* < 0.05) increase in mice lifespan compared to control.

**Figure 7 microorganisms-13-00076-f007:**
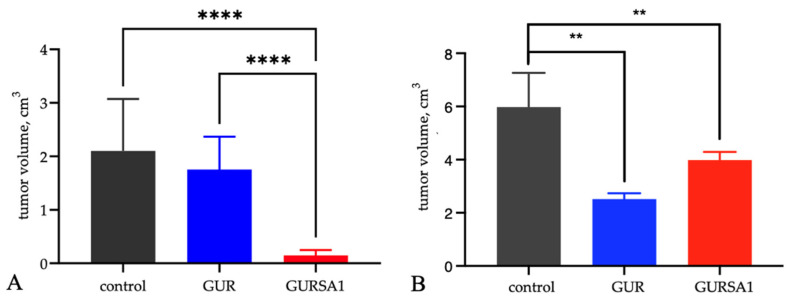
Influence of *S. pyogenes* strains on tumor volume in C57BL/6 mice with (**A**) murine hepatoma and (**B**) pancreatic cancer was statistically significant (** *p* < 0.01, **** *p* < 0.0001) 10 days following inoculation with *S. pyogenes* GUR and *S. pyogenes* GURSA1.

**Table 1 microorganisms-13-00076-t001:** Contamination of mice tumors, livers, and spleens within one day after intraperitoneal injection of *S. pyogenes* GURSA1 strain.

Sample	Tumor, LgCFU	Spleen, LgCFU	Liver, LgCFU
GURSA1 (*n* = 6)	4.50 ± 1.38	0	0
Control (*n* = 6)	0	0	0

**Table 2 microorganisms-13-00076-t002:** Body weight changes in mice administered *S. pyogenes* strains.

Strains		Control	*S. pyogenes* GURSA1		*S. pyogenes* GUR	
Dose, CFU		0	10^6^	10^8^	10^6^	10^8^
Body weigth, g	Day 1	20.7 ± 0.6	20.5 ± 0.5	21 ± 0.7	20.5 ± 0.6	20.2 ± 0.5
	Day 7	20.5 ± 0.5	20.4 ± 0.4	20.3 ± 0.3	20.0 ± 0.5	19.9 ± 0.6
	Day 14	20.8 ± 0.5	20.6 ± 0.5	20.6 ± 0.9	20.0 ± 0.1	20.4 ± 1.4

## Data Availability

All data generated in this study are available within the article, the raw data are available at https://figshare.com/s/807fdbe107a4b75471f2, accessed on 13 December 2024.

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
