# Peer review of "Studying the Oncolytic Activity of *Streptococcus pyogenes* Strains Against Hepatoma, Glioma, and Pancreatic Cancer *In Vitro* and *In Vivo"

_microorganisms, 2025, doi:10.3390/microorganisms13010076_

Round 1

Reviewer 1 Report

Comments and Suggestions for Authors

Manuscript submitted to Microoganisms

Article Type:

Experimental.

Title:

Studying the Oncolytic Activity of Streptococcus pyogenes Strains against Hepatoma, Glioma, and Pancreatic Cancer in vitro and in vivo

Section:

Medical Microbiology

_______

Dear Editor,

Thank you very much for the opportunity to review this manuscript submitted to Microorganisms.

Please see my few suggestions below that can help improve the quality of the manuscript.

_______

TITLE

The title is adequate.

_______

OVERALL COMMENTS

A)  According to the information that cancer is a leading cause of mortality globally, there is a need to amplify therapeutic approaches. Conventional treatment modalities, include radiation and chemotherapy, not always lead to complete remission, highlighting the critical need for novel therapeutic strategies. Group A Streptococcus (GAS) strains has oncolytic potential for tumor treatment. In this sense, the authors intended to investigate the oncolytic efficacy of S. pyogenes GUR and its M protein knockout mutant, S. The hepatoma 22A, pancreatic cancer PANC02, and human glioma U251 cells, both in vitro and in vivo, using the C57BL/6 mouse model. Methods: The in vitro oncolytic cytotoxic activity of GAS strains was studied against human glioma U251, pancreatic cancer PANC02, murine hepatoma 22a, and normal skin fibroblast cells using the MTT assay and the real-time xCELLigence system. The main results showed that that tumor treatment with S. pyogenes GURSA1 significantly increased the lifespan of C57BL/6 mice with hepatoma (34 days, p=0.040) and pancreatic cancer (32 days, p=0.039) relative to the control groups (24 and 28 days, respectively). “The overall survival rate and lifespan of mice treated with S. pyogenes GURSA1, a strain lacking the M protein on its surface, were significantly higher compared to the control and S. pyogenes GUR strain groups”.

B)   I suggest including newer references along with the text.

There are some good studies that can be found at PUBMED or Google Scholar databases.

C)   Figures are fine but Figure 2 is too small. I suggest that the authors include A, B, and C in separate Figures or in larger scale.

D)  In some parts of the text, S. pyogenes is not in italics. Please check along with the entire text. The same fo in vitro and in vivo (see as an example line 419).

_______

ABSTRACT

          This section is adequate.

_______

KEYWORDS

          Keywords are also adequate.

_______

INTRODUCTION

There is a need to include newer references in this section. So many articles published in 2023 and 2024.

Please check PUBMED, EMBASE, Cochrane, and Google Scholar.

_______             

METHODS

In lines 263-268 we can read that:

“2.9. Ethics statement 264 All animal experiments were conducted in accordance with the “Rules of Laboratory 265 Practice” (Ministry of Health of the Russian Federation, No. 708). The study was ap- 266 proved by the Local Ethics Committee for Animal Care and Use at the Institute of Ex- 267 perimental Medicine, Saint Petersburg, Russia.”

I suggest including the Ethical concerns in the beginning of Methods section.

RESULTS

This section is well performed. Just one comment. In lines 383-389, we can read that “3.4. Studying of acute oral toxicity of S. pyogenes GUR and GURSA1 strains Acute oral toxicity was assessed using a two-dose acute oral toxicity test. The doses  were selected based on the potential therapeutic dose — 106 CFU per mouse — and a 100-fold higher dose, 108CFU per mouse. No fatalities or clinical signs were observed in the mice throughout the study. There were no significant changes in body weight in the  rats administered S. pyogenes strains (Table 2). Furthermore, no abnormal findings were noted during necropsy. These results suggest that S. pyogenes GUR and S. pyogenes GURSA1 did not adversely affect the health of the mice.”

Table 2 is mentioned here but it is not in the text.

CONCLUSION

This section is adequate. However, I suggest including the limitations of this study

_______

REFERENCES

          There is a need to include newer references in the manuscript.

Author Response

Dear Reviewer,

Thank you for your time and thoughtful feedback. Please find our responses attached for your review.

Sincerely,
The Authors

Reviewer 2 Report

Comments and Suggestions for Authors

Dear Authors, 

The manuscript titled "Studying the Oncolytic Activity of Streptococcus Pyogenes Strains against Hepatoma, Glioma, and Pancreatic Cancer in vitro and in vivo" highlights the need for innovative cancer therapies. This study examines the oncolytic potential of Streptococcus pyogenes strains GUR and its M protein-deficient mutant, GURSA1, on murine hepatoma (22A), pancreatic cancer (PANC02), and human glioma (U251) cells.

GURSA1 showed strong activity against U251 cells (86.5%) and moderate effects on 22A (36.5%), while sparing healthy fibroblasts. In mouse models, it increased survival and slowed tumor progression, making it a promising candidate for novel cancer therapies.

During the review, I highlighted the following critical points

Introduction

  • Background Context: The introduction provides extensive historical context but risks overwhelming the reader with too much detail before presenting the study's objective.
  • Focus: Narrowing the scope to the most relevant historical milestones could improve readability.
  • Objective Statement: The aim of the study is clear but could be stated more succinctly to emphasize its novelty.

Methods

  • Detail: The methodology is described comprehensively; however, some steps (e.g., cell culture protocols and statistical analyses) include unnecessary detail that could be condensed.
  • Reproducibility: The level of detail supports reproducibility but may challenge readability. Consider moving highly technical details to supplementary materials.
  • Organization: The section could benefit from better organization, grouping similar methods (e.g., cell assays and animal models) to streamline the narrative.

Results

  • Figures and Tables: The integration of figures and tables is effective but could be accompanied by more concise descriptions in the text to avoid redundancy.
  • Interpretation: While statistical results are provided, the biological significance could be more explicitly interpreted alongside the data.
  • Clarity: Some comparisons between strains (e.g., GUR vs. GURSA1) could be presented more clearly, emphasizing their clinical relevance.

Discussion

  • Depth of Analysis: The discussion provides valuable insights but could more thoroughly connect findings to broader clinical implications and existing literature.
  • Critical Reflection: A more explicit acknowledgment of the study's limitations and potential biases would strengthen the credibility of the conclusions.
  • Future Directions: These are mentioned but could be expanded to provide a clearer roadmap for subsequent research.

Conclusion

  • Impact: The conclusion summarizes findings well but could more strongly highlight the significance of GURSA1 as a therapeutic candidate.
  • Conciseness: Reducing overlap with the discussion section would improve readability.

References

  • Currency: Most references are recent and relevant; however, ensuring all are the latest available versions strengthens the paper.
  • Diversity: Including a broader range of studies could enhance the paper's perspective.

Author Response

(The authors gave the same response as above.)

Reviewer 3 Report

Comments and Suggestions for Authors

The manuscript "Studying the Oncolytic Activity of Streptococcus Pyogenes Strains against Hepatoma, Glioma, and Pancreatic Cancer in vitro and in vivo" is a fascinating and innovative exploration of bacterial-mediated cancer therapy. The authors investigate the therapeutic potential of two S. pyogenes strains: the wild-type GUR and the genetically modified GURSA1 strain, which lacks the M protein. This approach addresses a significant gap in cancer treatment by offering an alternative strategy to conventional therapies, particularly for resistant and metastatic cancers. The study is methodologically robust, combining in vitro cytotoxicity assays, including MTT and RT xCELLigence, with well-designed in vivo mouse models. The genetic engineering of the GURSA1 strain represents a notable advancement, reducing the strain's virulence while retaining its oncolytic properties. The comprehensive safety evaluation, including the lack of toxicity on normal human fibroblasts and minimal adverse effects in animal models, adds significant weight to the findings.The introduction effectively contextualizes the work within the broader scope of cancer therapy and the history of bacterial applications in oncology. However, it could be enhanced with more recent references to underscore the relevance of the study in the current scientific landscape. The authors provide a clear rationale for targeting glioma, pancreatic cancer, and hepatoma cells, although a more detailed explanation of the selection criteria for these models would strengthen the argument. The methods are described in commendable detail, demonstrating a high level of technical expertise. However, visual aids, such as plasmid maps or gel electrophoresis results confirming the genetic modifications, could enhance the clarity and accessibility of this section.The results are presented with clarity and supported by statistical analyses, including Student's t-tests and Mann-Whitney U-tests, ensuring the reliability of the data. Figures and graphs, while informative, could benefit from higher resolution and more detailed captions to highlight key findings. For instance, the addition of annotations on tumor growth and survival curves would improve the visual communication of the results. The discussion provides an insightful interpretation of the data, linking the findings to the broader implications for microbial cancer therapy. A deeper exploration of the potential clinical applications, including the selective tumor colonization observed with the GURSA1 strain, would enrich the narrative. Furthermore, addressing potential challenges in translating these findings to human trials, such as tumor heterogeneity or immune system interactions, would provide a balanced perspective.

The conclusion succinctly summarizes the study's contributions, emphasizing the therapeutic potential of GURSA1 in extending survival and inhibiting tumor growth. However, a more explicit acknowledgment of the study's limitations, such as the focus on preclinical models or the need for long-term toxicity studies, would add depth. The manuscript is a significant contribution to the field of bacterial-mediated cancer therapy, showcasing both technical innovation and therapeutic promise. With minor revisions to enhance the discussion, improve figure quality, and expand on clinical implications, the study could serve as a valuable reference for future research and development in this emerging field. I recommend the manuscript for publication after addressing these minor adjustments.

Author Response

(The authors gave the same response as above.)
